# Three Pairs of Novel Enantiomeric 8-*O*-4′ Type Neolignans from *Saussurea medusa* and Their Anti-inflammatory Effects In Vitro

**DOI:** 10.3390/ijms232214062

**Published:** 2022-11-15

**Authors:** Jing-Ya Cao, Qi Dong, Zhi-Yao Wang, Li-Juan Mei, Yan-Duo Tao, Rui-Tao Yu

**Affiliations:** 1Qinghai Provincial Key Laboratory of Tibetan Medicine Research, Northwest Institute of Plateau Biology, Chinese Academy of Sciences, Xining 810008, China; 2University of Chinese Academy of Sciences, Beijing 100049, China; 3Henan Academy of Science, Zhengzhou 450002, China

**Keywords:** *Saussurea medusa*, neolignans, anti-inflammatory activity, ECD calculation, molecular docking, iNOS protein

## Abstract

Three pairs of novel enantiomeric 8-*O*-4′ type neolignans (**1a**/**1b**–**3a**/**3b**), together with seven known analogues (**4**–**10**), were isolated from the whole plants of *Saussurea medusa.* Their structures were elucidated by extensive spectroscopic data analysis and electric circular dichroism (ECD) calculations after chiral separations. All compounds were obtained from *S. medusa* for the first time, and compounds **1**–**3** and **5**–**10** had never been obtained from the genus *Saussurea* previously. The anti-inflammatory activities of the compounds were evaluated by determining their inhibitory activities on the production of NO and inducible nitric oxide synthase (iNOS) expression in LPS-stimulated RAW 264.7 cells. Compounds (+)-**1a**, (−)-**1b** and **5**–**7** inhibited NO production and had IC_50_ values ranging from 14.3 ± 1.6 to 41.4 ± 3.1 μM. Compound **7** induced a dose-dependent reduction in the expression of iNOS in LPS-treated RAW 264.7 cells. Molecular docking experiments showed that all active compounds exhibited excellent docking scores (<−7.0 kcal/mol) with iNOS. Therefore, compounds (+)-**1a**, (−)-**1b** and **5**–**7** isolated from the whole plants of *S. medusa* may have therapeutic potential in inflammatory diseases.

## 1. Introduction

The Asteraceae plant, *Saussurea medusa* Maxim., is a rare perennial medicinal herb that grows in the northwestern part of China (e.g., Tibet, Xinjiang, Qinghai, Gansu, Yunnan and Sichuan provinces) at heights of 3500–5300 m [1]. *S. medusa* has been commonly used for the treatment of rheumatic arthritis, menoxenia, gynopathy, traumatic bleeding, anthrax, febrile tingling and headache [2,3]. Some phytochemical studies on this plant regarding flavonoids, lignans, terpenoids and chlorophyll constituents [1,4,5,6] have been previously reported. These compounds exhibit an array of biological activities, such as anti-inflammatory [7], antitumor [8] and immunosuppressive [4] activities. In our previous study, a series of arylnaphthalene lignans were isolated from *S. medusa*, and some of them exhibited anti-inflammatory activity, which prompted us to continue researching this plant [9].

As part of our ongoing projects on searching for novel bioactive compounds from *S. medusa*, we presented the isolation of three pairs of novel enantiomeric 8-*O*-4′ type neolignans (**1a**/**1b**–**3a**/**3b**), together with seven known analogues (**4**–**10**), from the whole plants of *S. medusa*. Their structures were elucidated based on extensive spectroscopic data and time-dependent density-functional-theory-based electronic circular dichroism (TDDFT-ECD) calculations [10] after chiral separations.

Enantiomers usually exist in the form of racemates or partial racemates in nature. Although high-performance liquid chromatography (HPLC), electrophoresis dominated chiral separation and other various separation techniques have been developed to separate enantiomers [11], separation still remains a challenge due to their similar chemical and physical properties in a chiral environment [12]. However, it is well known that enantiomers differ qualitatively or quantitatively in their pharmacological, toxicological, and biological activities [13]. Therefore, it is necessary to obtain optically pure compounds and evaluate their pharmacological effects.

Inflammation is a central feature of many pathophysiological conditions in response to tissue injury and host defenses against invading microbes [14]. The pathogenesis of many diseases, including cancer [15], diabetes [16,17], neurodegenerative [18], Parkinson’s disease [19], cardiovascular [20,21] and other life-threatening diseases involve inflammation. Usually, the occurrence of inflammation is accompanied by the activation of various immune cells, especially macrophages, which participate in the initiation and spread of the inflammatory response by releasing proinflammatory cytokines and mediators such as endogenous radical nitric oxide (NO) [22]. NO is generated by NO synthases (NOSs) through the oxidation of *L*-arginine to *L*-citrulline. Two constitutive isoforms (cNOS) are detected in neuronal tissues (nNOS) and vascular endothelial cells (eNOS), whereas inducible NOS (iNOS) is expressed in various cell types (e.g., macrophages) upon inflammatory stimulation [23]. Inhibiting the activity of iNOS to block excessive NO production has been considered a promising strategy for the treatment of inflammatory diseases [24]. Therefore, the anti-inflammatory activities of the compounds were preliminarily evaluated in vitro by examining their ability to inhibit LPS-induced NO production and iNOS expression in RAW 264.7 macrophage-like cells.

In silico approaches such as molecular docking are considered one of the fundamental elements of drug design and discovery paradigms aimed at elucidating ligand–receptor interaction mechanisms and assisting lead optimization [25]. To preliminarily explore the anti-inflammatory mechanism, molecular docking experiments were performed to examine the interactions between the active compounds and the iNOS protein.

## 2. Results

### 2.1. Isolation and Identification of Compounds

The contents of *S. medusa* were extracted with 95% ethanol to give a crude extract, which was then suspended in H_2_O and successively partitioned with petroleum ether, EtOAc and *n*-butanol. The EtOAc fraction was separated by various column chromatographic methods to afford three new pairs of enantiomeric lignans (**1a**/**1b**–**3a**/**3b**) and seven known analogues (**4**–**10**) (Figure 1).

Medusidine A (**1**) had a molecular formula of C_22_H_30_O_8_, as deduced from NMR data and a sodium adduct ion at *m/z* 445.1822 [M + Na]^+^ (calcd for C_22_H_30_NaO_8_, 445.1833) in the (+)-HRESIMS. The IR spectrum of **1** displayed characteristic absorption peaks of hydroxy (3447 cm^–1^), carbonyl (1711 cm^–1^) and aromatic ring (1589, 1517 and 1025 cm^–1^) groups. The ^1^H NMR spectrum (Table 1) showed two sets of 1,2,3,5-tetrasubstituted aromatic rings with two equivalent aromatic protons at *δ*_H_ 6.52 (2H, s, H-2, 6) and 6.44 (2H, s, H-2′, 6′). A 2,3-propane-diol moiety was indicated by signals of one oxymethine at *δ*_H_ 4.18 (1H, m, H-8), one methylene at *δ*_H_ 3.20 (1H, dd, *J* = 13.6, 5.4 Hz, H-7a), 2.97 (1H, dd, *J* = 13.6, 8.6 Hz, H-7b) and an oxymethylene at *δ*_H_ 3.56 (1H, dd, *J* = 12.4, 2.4 Hz, H-9a), 3.43 (1H, dd, *J* = 12.4, 3.8 Hz, H-9b). Meanwhile, a 3-propanol moiety was indicated by signals at *δ*_H_ 2.66 (2H, m, H-7′), 1.88 (2H, m, H-8′) and 3.69 (2H, t, *J* = 6.4 Hz, H-9′). Additionally, four methoxy groups attached to the aromatic ring at *δ*_H_ 3.87 (6H, s, 3,5-O*Me*) and 3.83 (6H, s, 3′,5′-O*Me*) were also observed. The ^13^C NMR spectrum (Table 1) of **1** revealed 22 carbons, which were resolved into 12 aromatic carbons arising from two benzene rings, five methylene carbons (two oxygenated), one oxygenated methine and four methoxy carbons.

The HMBC correlations (Figure 2) of H_2_-7/C-1, C-2 and C-6; H_2_-7′/C-1′, C-2′ and C-6′, in combination with the ^1^H-^1^H COSY correlations (Figure 2) of H_2_-7/H-8/H_2_-9 and H_2_-7′/H_2_-8′/H_2_-9′, confirmed the presence of two phenyl propanoid units. These two phenyl propanoid groups were linked by the formation of an ether bond between C-8 and C-4′, and this deduction was confirmed by the HMBC correlation of H-8 to C-4′. These NMR spectroscopic data suggested **1** to be an 8-*O*-4′ system neolignan, which closely resembled that of 3,3′,5,5′-tetramethoxy-8,4′-oxyneolignane-4,9,9′-triol-4-*O*-*β*-D- glucopyranoside [26]. The only difference between them was the absence of C1′′-C6′′ in **1**, which was in accordance with its molecular formula, suggesting that compound **1** was the aglycone of the known compound. However, the absolute configuration of 3,3′,5,5′-tetramethoxy-8,4′-oxyneolignane-4,9,9′-triol-4-*O*-*β*-D-glucopyranoside had not been determined. Herein, ECD calculations were used to determine the absolute configuration of **1**.

Compound **1** showed weak CE in the ECD spectrum and immeasurable optical rotations, which implied that **1** was likely a racemic mixture. This prediction was confirmed by the presence of two peaks in chiral HPLC analysis using a Daicel IG column. Compounds (+)-**1a** and (−)-**1b** were successfully separated in a ratio of approximately 1:1 (see Appendix A). By comparing their calculated ECD and experimental ECD (Figure 3), the calculated ECD curve of the (8*S*) form matched well with the experimental ECD spectrum of (−)-**1b** (Figure 3), which allowed the determination of the absolute configuration of (−)-**1b** as 8*S*. Thus, the almost symmetrical ECD curve of its enantiomer (+)-**1a** was assigned as 8*R.*

Medusidine B (**2**) was determined to be C_22_H_30_O_7_ from the (+)-HRESIMS ion peak at *m/z* 429.1896 [M + Na]^+^ (calcd for C_22_H_30_NaO_7_, 429.1884). The ^1^H NMR and ^13^C NMR spectra (Table 1) of **2** showed two sets of 1,2,4-trisubstituted aromatic rings, five methylene carbons (three oxygenated), two oxygenated methines, two methoxy groups and one upfield methyl group. The aforementioned NMR features suggested its structure to be closely related to that of **1**, indicating that they were structural analogs. The differences were the absence of 5/5′-O*Me* and the existence of an ethyoxy unit at C-7 in **2**. This was evident from the upfield chemical shifts of C-5 (*δ*_C_ 115.6) and C-5′ (*δ*_C_ 119.4), as verified by the ^1^H–^1^H COSY correlations (Figure 2) of H-5/H-6 and H-5′/H-6′. Furthermore, the presence of an ethyoxy group at C-7 in **2** was established based on the downfield chemical shifts of C-7 (*δ*_C_ 81.8) in combination with the HMBC correlation (Figure 2) from H-7 to C-1″, which was further verified by the ^1^H–^1^H COSY correlation of H_2_-1″/H_3_-2″.

In terms of the possible staggered conformers with intramolecular hydrogen bonding of the benzylic hydroxy and aryloxy groups, the large and small *J* values for H-7 and H-8 of 8-*O*-4′ neolignan diastereoisomers correspond to the *threo* form and *erythro* form, respectively [27]. In the ^1^H NMR spectrum of **2** in CD_3_OD, a large coupling constant (*J*_7, 8_ = 6.4 Hz) was observed. Thus, the relative configuration of C-7 and C-8 was deduced to be in the *threo* form. Compound **2** showed weak Cotton effects in its ECD spectrum and immeasurable optical rotations, and its partially racemic nature was confirmed by chiral HPLC analysis in the same manner as that of **1**. Compounds (−)-**2a** and (+)-**2b** were obtained with a variable enantiomeric excess (ee) value of approximately 40% for (+)-**2b** (see Appendix A). The negative CE at 239 nm in the ECD spectrum of (−)-**2a** indicated a 7*R*,8*R* configuration, while (+)-**2b** was deduced to be a 7*S*,8*S* configuration due to the positive CE at 239 nm [28,29]. Furthermore, ECD calculations also supported their absolute configurations.

Medusidine C (**3**) displayed a quasi-molecular ion peak at *m/z* 429.1887 (calcd for C_22_H_30_NaO_7_, 429.1884) in the (+)-HRESIMS analysis, corresponding to a molecular formula of C_22_H_30_O_7_. The IR, UV and NMR spectroscopic data of **3** were highly similar to those of **2**, suggesting that the planar structure of **3** was the same as that of **2**. The small coupling constant (*J*_7, 8_ = 5.9 Hz) in the ^1^H NMR spectrum of **3** suggested a relative-*erythro* configuration.

The ECD and optical rotation data indicated that **3** was also a pair of enantiomers. This deduction was supported by two peaks [ee = 38% for (+)-**3a**, see Appendix A] observed in the chiral HPLC analysis. Subsequently, chiral resolution was carried out to prepare optically pure (+)-**3a** and (−)-**3b**. Meanwhile, the positive CE at 239 nm of (+)-**3a** justified a 7*R*,8*S* configuration and the negative CE at 239 nm of (−)-**3b** justified a 7*S*,8*R* configuration. This was also supported by the ECD calculations.

Along with the new lignans, seven known analogues, (7*S*,8*R*)-4,7,9,9′-tetrahydroxy-3,3′-dimethoxy-8-*O*-4′-neolignan (**4**) [29]; (7*S*,8*R*)-syringylglycerol-8-*O*-4′-(synapylal cohol) ether (**5**) [30]; *threo*-guaiacylglycerol-8-*O*-4′-sinapylether (**6**) [31]; (−)-(7*R*,7′*R*,7″*R*, 8*S*,8′*S*,8″*S*)-4′,4″-dihydroxy-3,3′,3″,5-tetramethoxy-7,9′:7′,9-diepoxy-4,8″-oxy-8,8′-sesquineolignan-7″,9″-diol (**7**) [32]; (−)-(7*R*,7′*R*,7″*R*,8*S*,8′*S*,8″*S*)-4′,4″-dihydroxy-3,3′,3″,5,5′,5″-hexa methoxy-7,9′:7′,9-diepoxy-4,8″-oxy-8,8′-sesquineolignan-7″,9″-diol (**8**) [32]; (−)-(7*R*,7′*R*,7″*S*, 8*S*,8′*S*,8″*S*)-4′,4″-dihydroxy-3,3′,3″,5,5′,5″-hexamethoxy-7,9′:7′,9-diepoxy-4,8″-oxy-8,8′-sesquineolignan-7″,9″-diol (**9**) [32] and firmianols B (**10**) [33], were also obtained and identified on the basis of spectroscopic analysis and comparison with the reported literature.

### 2.2. Anti-Inflammatory Effects

All isolated compounds were evaluated for cell viability in RAW 264.7 macrophages. The results revealed that none of the compounds displayed cytotoxicity at the measured concentrations. Subsequently, all isolated constituents were screened for their inhibitory effects on NO production in LPS-induced RAW 264.7 macrophages. Quercetin was selected as a positive control with an IC_50_ value of 15.9 ± 1.2 μM. The results showed (Table 2) that compound **7** exhibited obvious suppressive activity on the production of NO with an IC_50_ value of 14.3 ± 1.6 μM, which was comparable to that of the positive control quercetin. Compounds (+)-**1a**, (−)-**1b**, **5** and **6** displayed moderate inhibitory activities with IC_50_ values ranging from 18.5 ± 1.9 to 41.4 ± 3.1 μM.

As far as we know, the most commonly recorded lignans in *S. medusa* are dibenzylbutyrolactone and tetrahydrofuran lignans [1,3,4]. In our previous study, a series of arylnaphthalene lignans with anti-inflammatory activities were isolated from *S. medusa* [9]. To date, however, 8-*O*-4′ neolignans have not been reported in *S. medusa*. As a major class of lignans, 8-*O*-4′ neolignans have been reported to have a wide range of bioactivities, especially in inflammatory responses [33,34,35]. In the present study, 8-*O*-4′ neolignans (compounds **1a**, **1b**, **5** and **6**) and sesquilignan (compound **7**) isolated from the whole plants of *S. medusa* significantly inhibited NO production in LPS-stimulated Raw 264.7 macrophages. To our knowledge, this is the first time that the anti-inflammatory effects of compounds **1**–**3** and **5** have been reported. In addition, compounds **6** and **7**, which had been isolated from the stems of *Firmiana simplex* [34] and the leaves and twigs of *E. alatus* [35], respectively, showed comparatively significant inhibitory effects as reported in the previous literature. The results of this study also further confirmed their anti-inflammatory activities.

Some preliminary structure–activity relationships could be drawn. The novel enantiomeric (+)-**1a** and (−)-**1b** exhibited similar anti-inflammatory activities. The absence of 5/5′-O*Me* and the presence of an ethyoxy group at C-7 weakened the inhibitory activities (compounds **2** and **3**). The C7′-C8′ double bond (compounds **5** and **6**) was found to be essential for the observed inhibitory effects. The absence of the C7′-C8′ double bond resulted in a loss of activity, as compound **4**, which lacked this property, displayed poor inhibitory activity. Compound **7** exhibited high activity mostly due to its planar structure and stereoselectivity. The introduction of a hydroxy group in the ditetrahydrofuran ring led to a loss of activity (compound **10**). Compounds **8** and **9** were inactive, likely due to the additional 5′/5″-O*Me* groups on aromatic rings.

### 2.3. Effect of the Selected Active Compounds on iNOS Expression

The NO inhibition results showed that compound **7** exhibited comparable activity to the positive control quercetin. Considering these results, compound **7**, with a relatively high residual amount (11 mg), was selected for assessment of iNOS protein expression by western blot. As demonstrated in Figure 4, iNOS protein expression was significantly increased following stimulation with LPS, and compound **7** caused a dose-dependent reduction in expression of iNOS in LPS-treated RAW 264.7 cells. The results revealed that compound **7** inhibited the production of NO by reducing iNOS protein expression.

### 2.4. Molecular Docking Studies

To explore the possible mechanism of inhibiting NO production and iNOS protein expression, molecular docking studies were performed to investigate the interactions between active compounds and the iNOS protein. The bioactive compounds (+)-**1a**, (−)-**1b**, **5**–**7** and positive control quercetin were selected for molecular docking studies. Table 3 summarizes the binding affinity and binding interactions of active compounds with iNOS. Through careful analysis of the results of NO inhibition and molecular docking experiments, it was discovered that the actual NO inhibition effects of these active compounds corresponded well with the molecular docking results, except in the case of the positive control quercetin. Specifically, the orders of IC_50_ in the NO inhibition studies are as follows: compound **7** < quercetin < compound **6** < compound (−)-**1b** < compound (+)-**1a** < compound **5** (Table 2). The orders of minimal binding energies in the case of molecular docking studies are as follows: compound **7** < compound **6** < compound (+)-**1a** < compound (−)-**1b** = compound **5** < quercetin (Table 3). Compound **7** exhibited the lowest (−9.4 kcal/mol) docking score with iNOS, consistent with its strongest NO inhibition effect. The novel enantiomeric (+)-**1a** and (−)-**1b** showed similar docking scores (−7.8 and −7.7 kcal/mol), which was also highly consistent with their similar moderate NO inhibition effects.

The results of the three-dimensional (3D) molecular docking images of the target compounds are shown in Figure 5. The visualization results showed that hydrogen bonds and hydrophobic interactions were formed between target compounds and crucial amino acid residues of the iNOS protein. The docking results (Table 3) showed that all active compounds were anchored to the catalytic site of the iNOS protein through various bonds and exhibited excellent docking scores (<−7.0 kcal/mol) with the iNOS protein. Among them, compound **7** exhibited the strongest interactions with iNOS and had a free binding energy of −9.4 kcal/mol. The 3D diagram (Figure 5E) illustrated that compound **7** interacted with amino acid residues TYR367, TYR483, LEU203, PHE363, TRP457 and TRP188 of iNOS. Two hydrogen bonds were found between the hydroxy groups on the two terminal benzene rings of compound **7** and amino acid residues TYR367 and TYR483. These hydrogen bonds strengthened the interactions between compound **7** and iNOS. In addition, we observed *π*-*π* interactions between the terminal benzene ring of compound **7** and amino acid residue TRP188, suggesting that terminal benzene rings and hydroxy groups on them may play an important role in anti-inflammatory activity.

The above results provided a potential explanation for the mechanism by which active compounds inhibit NO production and iNOS protein expression. Moreover, the results indicated that molecular docking could be an important tool in drug discovery to accelerate the recognition and investigation of novel drug candidates.

## 3. Materials and Methods

### 3.1. General Experimental Procedures

Optical rotations (Na lamp, 589 nm) were determined in MeOH on a Rudolph Autopol VI automatic polarimeter. UV spectra were measured on a Shimadzu UV-2550 UV-visible spectrophotometer. ECD spectra were obtained with a JASCO J-815 spectrometer using a 0.1 cm path length sample cell. Chiral analysis experiments were carried out on a JASCO LC-J1500 instrument (equipped with an AS-4050 HPLC auto sampler, a PU-4185 binary, a CO-4060 column oven and an MD-4014 photo diode array detector). NMR spectra were recorded on a Bruker Avance III 600 MHz spectrometer (Bruker Biospin AG, Zurich, Switzerland) with TMS as an internal standard. IR spectra were acquired on a Thermo IS5 spectrometer with KBr panels. (±)-ESIMS and (±)-HRESIMS analyses were performed on a Bruker Daltonics Esquire 3000 Plus LC-MS instrument and a Waters Q-TOF Ultima mass spectrometer, respectively. Chromatographic silica gel (200–300 and 300–400 mesh, Qingdao Haiyang Chemical Co. Ltd., Qingdao, China), Sephadex LH-20 (GE Healthcare, Uppsala, Sweden) and MCI gel (CHP20P, 75–150 μm, Mitsubishi Chemical Industries, Ltd., Tokyo, Japan) were used for column chromatography (CC). Precoated silica gel GF254 plates (Qingdao Haiyang Chemical Co. Ltd.) were used for TLC detection. Semipreparative HPLC separations were performed on a Waters 2695 binary pump system equipped with a Waters 2489 detector (210 and 254 nm) using a Waters X-Bridge Prep C18 column (5 μm, 10 × 250 mm) or a YMC-Pack ODS-A column (5 μm, 10 × 250 mm). Daicel Chiralpak IH (5 μm, 4.6 × 250 mm) and Daicel Chiralpak IG (5 μm, 4.6 × 250 mm) columns were used for chiral HPLC separations. Solvents used for HPLC were of HPLC grade (J & K Scientific Ltd., Beijing, China) and other solvents were of analytical grade (Shanghai Chemical Reagents Co. Ltd., Shanghai, China).

### 3.2. Plant Material

The whole plants of *S. medusa* were collected from Yeniu Ditch (altitude 4100 m), Qilian County, Xining City, Qinghai Province, China, in August 2018 and were authenticated by Professor Lijuan Mei from Northwest Institute of Plateau Biology. A voucher specimen (access number: 0341202) was deposited at the Key Laboratory of Tibetan Medicine of the Chinese Academy of Sciences.

### 3.3. Extraction and Isolation

The air-dried and powdered whole herbs of *S. medusa* (15.0 kg) were soaked overnight and then extracted three times in 95% ethanol (12 h × 75 L) at 70 °C. The 95% ethanol extracts were concentrated under vacuum to afford a residue (800 g). The residue was then suspended in H_2_O (4 L) and successively partitioned with petroleum ether (5 × 4 L), EtOAc (5 × 4 L) and *n*-butanol (5 × 4 L). Based on TLC analysis, the EtOAc-soluble fraction (90 g) was fractionated into seven fractions F1−F7 using MCI gel column chromatography (5 × 40 cm, 100–200 mesh) eluted with MeOH-H_2_O (10% to 100%). Fraction F5 (26.4 g) was chromatographed on a silica gel column again eluting with CH_2_Cl_2_/MeOH (400:1 to 10:1) in gradient to afford seven fractions F5a−F5g. Fraction F5f (1.7 g) was successively separated via a Sephadex LH-20 column (3 × 150 cm) eluted with MeOH to afford subfractions F5f1−F5f7. Fraction F5f2 (343 mg) was further subjected to a silica gel column eluted with CH_2_Cl_2_ /MeOH (250:1 to 1:1) in gradient to give subfractions F5f21−F5f24. F5f22 (200 mg) was then separated over a Sephadex LH-20 column (2 × 150 cm) eluted with EtOH to afford subfractions F5f221−F5f223. F5f222 (59 mg) was then purified by semi-preparative HPLC with 46% MeOH in H_2_O as the mobile phase to afford compound **1** (13 mg, *t*_R_ = 29 min). Similarly, F5f221 (91 mg) was purified by semi-preparative HPLC with 44% MeOH in H_2_O as the mobile phase to yield **2** (6 mg, *t*_R_ = 48 min) and **3** (7 mg, *t*_R_ = 50 min). Fraction F5f3 (501 mg) was subjected to a silica gel column eluted with CH_2_Cl_2_/MeOH (150:1 to 1:1) in gradient to give subfractions F5f31−F5f38. F5f33 (112 mg) and F5f35 (67 mg) were fractioned via a Sephadex LH-20 column (2 × 150 cm) eluted with EtOH to afford subfractions F5f331−F5f333 and F5f351−F5f353, respectively. F5f332 (76 mg) was then purified by RP semi-preparative HPLC (44% MeOH in H_2_O) to yield **7** (11 mg, *t*_R_ = 14 min). F5f352 (29 mg) afforded **8** (4 mg, *t*_R_ = 23 min) and **9** (4 mg, *t*_R_ = 29 min) with 45% MeOH in H_2_O as mobile phase. Similarly, F5f36 (75 mg) afforded **10** (3 mg, *t*_R_ = 29 min) with 44% MeOH in H_2_O as mobile phase. Fraction F4 (15.8 g) was subjected to a silica gel column eluted with CH_2_Cl_2_/MeOH (400:1 to 1:1) in gradient to give subfractions F4a−F4k. Separation of F4k (1.0 g) with Sephadex LH-20 (MeOH) (3 × 150 cm) yielded subfractions F4k1−F4k3. Fraction F4k2 (243 mg) was subjected to a silica gel column eluted with *n*-hexane/isopropanol (80:1 to 1:1) in gradient to give subfractions F4k21−F4k23. F4k22 (97 mg) was then purified by RP semi-preparative HPLC (32% MeOH in H_2_O) to yield **3** (10 mg, *t*_R_ = 26 min), **5** (4 mg, *t*_R_ = 37 min) and **6** (4 mg, *t*_R_ = 41 min).

### 3.4. Details of New Compounds

#### 3.4.1. Medusidine A (**1**)

Light yellow gum; [*α*]^25^_D_ +3.3 (*c* 0.57 in MeOH); ^1^H and ^13^C NMR (CDCl_3_) data, see Table 1; IR (KBr) *ν*_max_ 3447, 2939, 1711, 1589, 1517, 1459, 1424, 1330, 1222, 1122, 1025 cm^−1^; UV (MeOH) *λ*_max_ (log *ε*) 208 (4.23), 236 (3.49), 277 (2.58) nm; (+)-ESIMS *m*/*z* 445.4 [M + Na]^+^; (−)-ESIMS *m*/*z* 421.3 [M − H]^−^; (+)-HRESIMS *m*/*z* 445.1822 [M + Na]^+^ (calcd for C_22_H_30_NaO_8_, 445.1833, Δ +2.35 ppm).

**1a**: light yellow gum; [*α*]^25^_D_ +67.7 (*c* 0.2 in MeOH); ECD (MeOH) λ (Δε) 246 (−2.76), 262 (−0.48), 277 (−0.79) nm;

**1b**: light yellow gum; [*α*]^25^_D_ −63.9 (*c* 0.2 in MeOH); ECD (MeOH) λ (Δε) 246 (+2.77), 262 (+0.63), 277 (+1.01) nm;

#### 3.4.2. Medusidine B (**2**)

Light yellow amorphous solid; [*α*]^25^_D_ +1.5 (*c* 0.20 in MeOH); ^1^H and ^13^C NMR (CD_3_OD) data, see Table 1; IR (KBr) *ν*_max_ 3406, 2931, 1709, 1604, 1511, 1454, 1270, 1225, 1155, 1125, 1035 cm^−1^; UV (MeOH) *λ*_max_ (log *ε*) 205 (3.90), 230 (3.31), 278 (2.95) nm; (+)-ESIMS *m*/*z* 429.3 [M + Na]^+^; (−)-ESIMS *m*/*z* 405.1 [M – H]^−^; (+)-HRESIMS *m*/*z* 429.1896 [M + Na]^+^ (calcd for C_22_H_30_NaO_7_, 429.1884, Δ –2.76 ppm).

**2a**: light yellow amorphous solid; [*α*]^25^_D_ –26.7 (*c* 0.1 in MeOH); ECD (MeOH) λ (Δε) 226 (+4.34), 239 (−2.11), 287 (−1.02) nm.

**2b**: light yellow amorphous solid; [*α*]^25^_D_ +27.9 (*c* 0.1 in MeOH); ECD (MeOH) λ (Δε) 226 (−3.71), 239 (+2.05), 287 (+1.15) nm.

#### 3.4.3. Medusidine C (**3**)

Light yellow gum; [*α*]^25^_D_ +4.0 (*c* 0.25 in MeOH); ^1^H and ^13^C NMR (CD_3_OD) data, see Table 1; IR (KBr) *ν*_max_ 3418, 2927, 1714, 1606, 1515, 1455, 1384, 1261, 1035 cm^−1^; UV (MeOH) *λ*_max_ (log *ε*) 207 (3.99), 230 (3.63), 281 (3.20) nm; (+)-ESIMS *m*/*z* 429.3 [M + Na]^+^; (−)-ESIMS *m*/*z* 405.0 [M − H]^+^; (+)-HRESIMS *m*/*z* 429.1887 [M + Na]^+^ (calcd for C_22_H_30_NaO_7_, 429.1884, Δ –0.78 ppm).

**3a**: light yellow gum; [*α*]^25^_D_ +63.5 (*c* 0.2 in MeOH); ECD (MeOH) λ (Δε) 226 (−4.37), 243 (+3.59), 275 (−0.12) nm.

**3b**: light yellow gum; [*α*]^25^_D_ –61.4 (*c* 0.2 in MeOH); ECD (MeOH) λ (Δε) 226 (+2.56), 243 (−2.24), 275 (+0.27) nm.

### 3.5. ECD Calculations

According to relevant literatures and program packages [36,37,38], the absolute configurations of **1a/1b**–**3a/3b** were determined by TDDFT-ECD calculations. For detailed calculation method, see Appendix A.

### 3.6. Determination of NO Production

Measurements of NO production in an activated macrophage-like cell line were performed as described previously [14]. Briefly, RAW 264.7 cells (purchased from Procell Life Science & Technology Co. Ltd., Wuhan, China) were cultured in plastic dishes in Dulbecco’s Modified Eagle Medium (DMEM) containing 10% fetal bovine serum (FBS) and supplemented with pyruvate (1.0 mM), glutamine (2.0 mM), streptomycin (10.0 μg/mL) and penicillin (100.0 U/mL). The cell lines were maintained at 37 °C in a humidified atmosphere with 5% CO_2_. The cells were treated with or without LPS (1.0 μg/mL) for 24 h in the presence or absence of the test compounds (3.125, 6.25, 12.5, 25.0 and 50.0 μM). Absorbance was measured at 540 nm after incubating culture supernatant (100 μL/well) with Griess reagent (100 μL) (Sigma-Aldrich, St. Louis, MO, USA) at room temperature, and the absorption coefficient was calibrated using a NaNO_2_ solution standard. Cell viability was measured using the MTT-based colorimetric assay according to a previous report [39].

### 3.7. Determination of iNOS Expression

As previously described in the literature [40], the treated cells were washed with PBS and suspended in lysis buffer. Cell debris was then removed after centrifugation. After the protein concentration was determined with BCA reagent, suspensions were boiled in SDS-PAGE loading buffer. The proteins were subjected to gel electrophoresis and electrophoretically transferred onto PVDF membranes (Millipore). The blot was incubated for 2 h with blocking solution at room temperature. After being washed, the membranes were incubated with a 1:1000 dilution of monoclonal anti-iNOS antibody and a 1:5000 dilution of *β*-actin antibody overnight at 4 °C. Blots were then washed three times with TBST and incubated with a 1:3000 dilution of secondary antibody solution for 1 h at room temperature. Blots were again washed three times with TBST and then detected by using enhanced chemiluminescence reagent and exposed to photographic films. Images were collected, and the related bands were quantitated by densitometric analysis using Gel-Pro analyzer software.

### 3.8. Molecular Docking Study

The specific docking methods and parameters can be seen in our previously published article [9]. Briefly, chemical structures of target compounds were drawn using ChemDraw 14.0 and converted to 3D coordinates in Chem3D. Each of them was subjected to energy minimization by the MM2 method and saved in “pdb” format. The 3D coordinates of the crystal structure of iNOS (PDB ID: 3E6T) were obtained from the RCSB Protein Data Bank (https://www.rcsb.org/pdb, accessed on 6 September 2022) [41] and handled in the Biovia Discovery Studio Visualizer 2020 program to check any missing residues/atoms and delete co-crystallized molecules such as cofactors, inhibitors, and water. The proteins and ligands were processed and converted to “pdbqt” format. A grid box with dimensions of 30, 30 and 30 points in x, y and z directions was built. Molecular docking was performed using AutoDock Vina 1.1.2 with default parameters, and the binding sites were defined within 10 Å around the co-crystallized ligands. Each docking involved nine independent runs. The docked model with the lowest docking energy was selected to represent its most favorable binding pattern.

## 4. Conclusions

In summary, three pairs of novel enantiomeric 8-*O*-4′ type neolignans (**1a**/**1b**–**3a**/**3b**), together with seven known analogues (**4**–**10**), were isolated from the whole plants of *S. medusa*. Their structures were established by spectroscopic data and ECD calculations. Compounds (+)-**1a**, (−)-**1b** and **5**–**7** displayed inhibitory activities on NO production with IC_50_ values ranging from 14.3 ± 1.6 to 41.4 ± 3.1 μM. Further iNOS protein expression studies demonstrated that compound **7** induced a dose-dependent reduction in iNOS protein expression. According to molecular docking studies, strong interactions were observed between active compounds and key residues of iNOS. Thus, a preliminary mechanism of inhibiting NO production and iNOS protein expression has been revealed. Overall, these findings not only extended the structural diversity of lignans in *S. medusa*, but also provided scientific background for the development of *S. medusa* as a potential medicinal plant to treat inflammatory diseases.

## Figures and Tables

**Figure 1 ijms-23-14062-f001:**
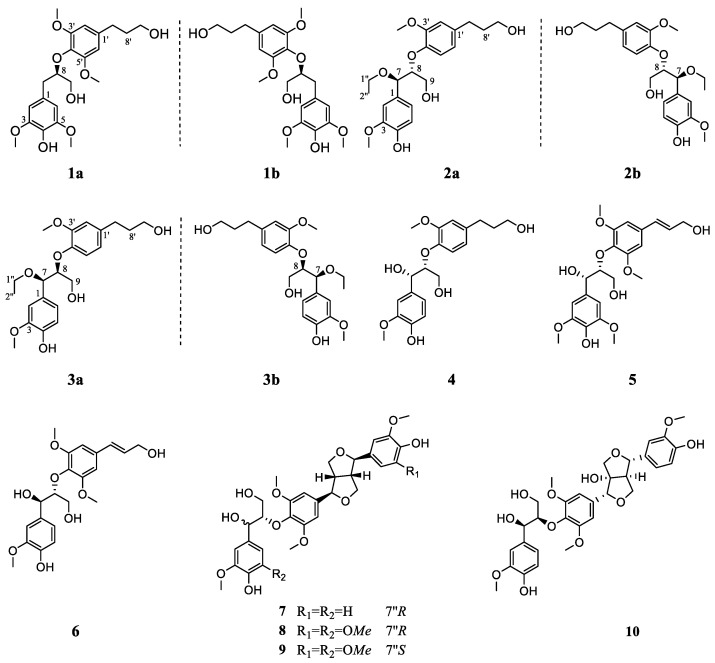
Chemical structures of compounds **1**–**10**.

**Figure 2 ijms-23-14062-f002:**
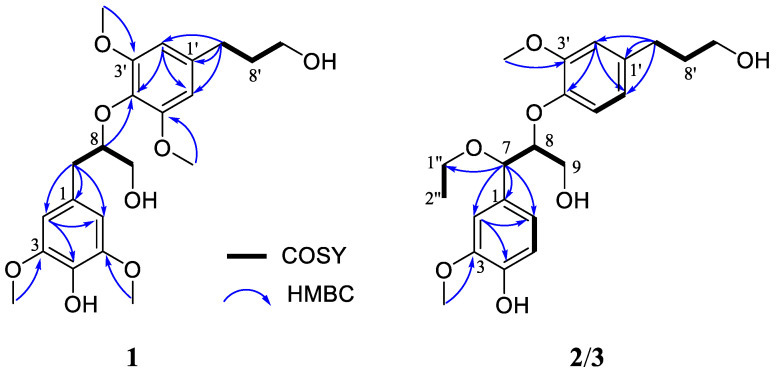
^1^H-^1^H COSY and key HMBC correlations of compounds **1**–**3**.

**Figure 3 ijms-23-14062-f003:**
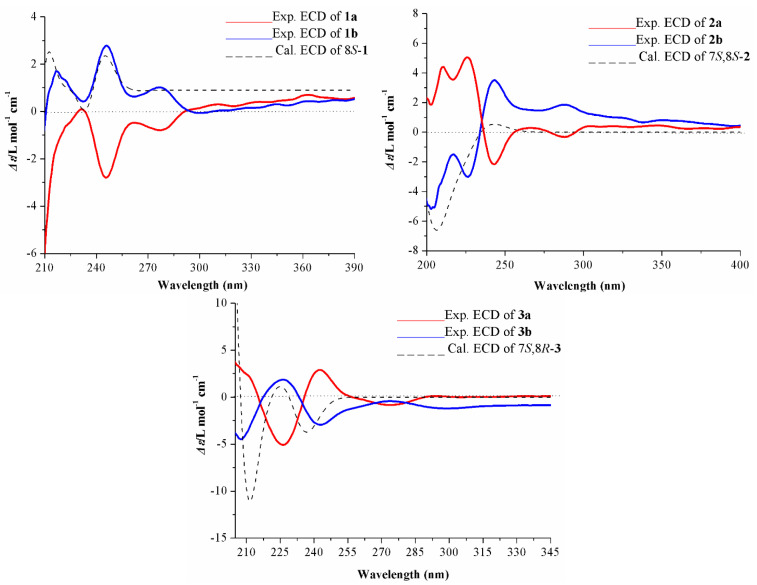
Experimental and calculated ECD spectra of compounds **1**–**3**.

**Figure 4 ijms-23-14062-f004:**
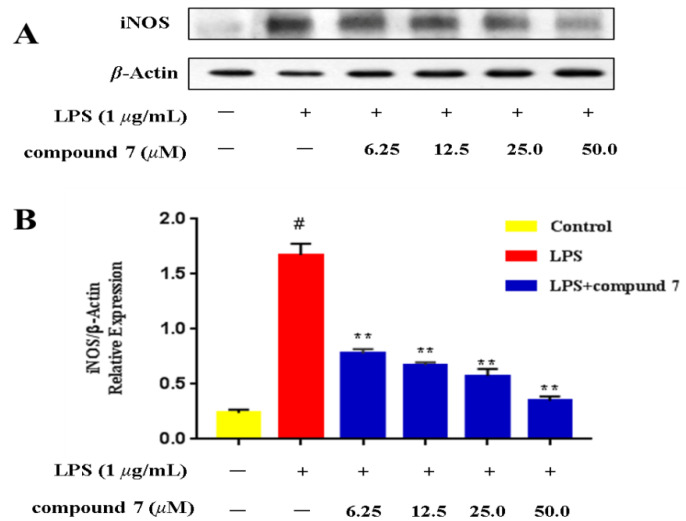
Concentration dependency of the inhibitory effects of compound 7 on iNOS enzyme. (**A**) Typical blotting of iNOS and *β*-actin. (**B**) The bar chart shows the quantitative evaluation of iNOS bands by densitometry. Data represent the mean ± SD (*n* = 3). ** *p* ≤ 0.01 compared with LPS-treated control. ^#^ LPS-treated control.

**Figure 5 ijms-23-14062-f005:**
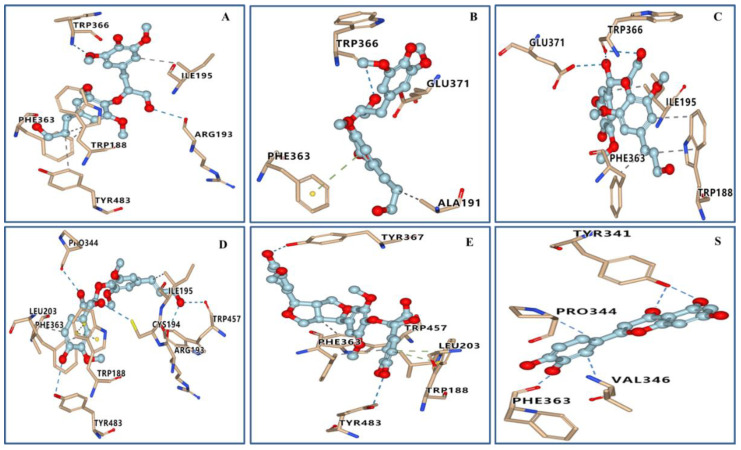
Molecular docking simulations of compounds **1a** (**A**), **1b** (**B**), **5** (**C**), **6** (**D**), **7** (**E**) and quercetin (**S**) with the iNOS protein.

**Table 1 ijms-23-14062-t001:** ^1^H NMR (400 MHz) and ^13^C (125 MHz) data for compounds **1**–**3** (*δ*_H_ in ppm, *J* in Hz).

Position	1 ^a^		2 ^b^		3 ^b^	
	*δ*_H_ (*J* in Hz)	δ_C_, Type	δ_H_ (*J* in Hz)	δ_C_, Type	δ_H_ (*J* in Hz)	δ_C_, Type
1	―	129.6, C	―	131.8, C	―	131.7, C
2	6.52, s	106.3, CH	6.90, d (1.8)	112.4, CH	6.97, d (1.7)	112.1, CH
3	―	147.0, C	―	148.8, C	―	149.0, C
4	―	133.3, C	―	147.3, C	―	147.4, C
5	―	147.0, C	6.72, d (8.1)	115.6, CH	6.77, d (8.0)	115.9, CH
6	6.52, s	106.3, CH	6.78, dd (8.1, 1.8)	122.0, CH	6.81, dd (8.0, 1.7)	121.4, CH
7	a 3.20, dd (13.6, 5.4)	38.0, CH_2_	4.46, d (6.4)	81.8, CH	4.53, d (5.9)	82.5, CH
	b 2.97, dd (13.6, 8.6)					
8	4.18, m	84.5, CH	4.27, ddd (6.4, 5.4, 3.5)	85.9, CH	4.26, ddd (5.9, 4.0, 3.5)	86.5, CH
9	a 3.56, dd (12.4, 2.4)	62.5, CH_2_	a 3.85, dd (11.9, 5.4)	62.3, CH_2_	a 3.63, dd (11.7, 4.0)	62.3, CH_2_
	b 3.43, dd (12.4, 3.8)		b 3.80, dd (11.9, 3.5)		b 3.42, dd (11.7, 3.5)	
1′	―	138.2, C	―	137.9, C	―	137.8, C
2′	6.44, s	105.7, CH	6.74, d (1.9)	114.1, CH	6.83, d (1.8)	114.0, CH
3′	―	153.3, C	―	151.7, C	―	151.7, C
4′	―	133.7, C	―	147.3, C	―	148.1, C
5′	―	153.3, C	6.69, d (8.2)	119.4, CH	6.92, d (8.2)	119.3, CH
6′	6.44, s	105.7, CH	6.61, dd (8.2, 1.9)	121.7, CH	6.69, dd (8.2, 1.8)	121.9, CH
7′	2.66, m	32.7, CH_2_	2.57, t (7.7)	32.6, CH_2_	2.62, t (7.7)	32.7, CH_2_
8′	1.88, m	34.4, CH_2_	1.76, m	35.5, CH_2_	1.81, m	35.6, CH_2_
9′	3.69, t (6.4)	62.3, CH_2_	3.53, t (6.5)	62.2, CH_2_	3.56, t (6.5)	62.2, CH_2_
1″			a 3.41, dq (9.3, 7.0)	65.4, CH_2_	a 3.43, dq (9.3, 7.0)	65.7, CH_2_
			b 3.38, dq (9.3, 7.0)		b 3.40, dq (9.3, 7.0)	
2″			1.15, t (7.0)	15.6, CH_3_	1.14, t (7.0)	15.6, CH_3_
OMe-3/3′	3.87, s/3.83, s	56.5,CH_3_/56.2,CH_3_	3.77, s/3.74, s	56.5,CH_3_/56.2,CH_3_	3.83, s/3.83, s	56.5,CH_3_/56.4,CH_3_
OMe-5/5′	3.87, s/3.83, s	56.5,CH_3_/56.2,CH_3_				

^a^ Measured in CDCl_3_. ^b^ Measured in CD_3_OD.

**Table 2 ijms-23-14062-t002:** Inhibition of LPS-induced NO production.

Compound	IC_50_ (μM) ^a^	Compound	IC_50_ (μM)
**1a**	26.4 ± 2.1	**5**	41.4 ± 3.1
**1b**	23.1 ± 1.8	**6**	18.5 ± 1.9
**2a**	>50	**7**	14.3 ± 1.6
**2b**	>50	**8**	>50
**3a**	>50	**9**	>50
**3b**	>50	**10**	>50
**4**	>50	Quercetin ^b^	15.9 ± 1.2

^a^ Data are expressed as the mean ± SD (*n* = 3). ^b^ Positive control.

**Table 3 ijms-23-14062-t003:** Docking results of active compounds with the iNOS protein.

Compound	Docking Scores (kcal/mol)	Hydrogen Bonds	Hydrophobic Interaction
**1a**	−7.8	TRP366, ARG193	TRP188, ILE195, PHE363, TYR483
**1b**	−7.7	TRP366, GLU371, ARG193, CYS194	ALA191
**5**	−7.7	TRP366, GLU371	TRP188, ILE195, PHE363
**6**	−8.5	PRO344, TRP457, TYR483	ILE195, LEU203, PHE363
**7**	−9.4	TYR367, TYR483	LEU203, PHE363, TRP457
quercetin	−7.5	TYR341, PHE363	PRO344, VAL346

## Data Availability

All data are contained within the article.

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
