# Peer review of "Three Pairs of Novel Enantiomeric 8-O-4′ Type Neolignans from Saussurea medusa and Their Anti-inflammatory Effects In Vitro"

_ijms, 2022, doi:10.3390/ijms232214062_

Round 1

Reviewer 1 Report

I believe that this is an interesting article towards the identification of new compounds and the unraveling of the potentials of Saussurea Medusa.

Some issues that need to be addressed and to be taken into account are listed below.

1. The “Isolation and identification of compounds” method has been standardized before? Why this system of solvents for extraction was used? In case of another extraction method would have been isolated different groups of compounds? Please add relevant reference in case there are. This is a very important part, since the article concerns Plant Sciences.

2. “The EtOAc fraction was separated by various column chromatographic methods” Are these the only isolated compounds or there are others that were not isolated and tested?

3. In the supplementary file the authors provide the spectrums of the 3 compounds. What about the other spectrums of the other known compounds?

4. “Along with the new lignans, three known analogues” In this sentence it seems all of them to be known analogues (compounds 4-10), and not the 3 of them.

5. The authors should explain how they concluded that these are the structures of compounds 4-10 as they did with compounds 1-3.

6. “whole plants of S. medusa have therapeutic potential in anti-inflammatory diseases” I think that since this is a preliminary study on one target (NO-iNOS) concerning inflammatory processes, and there are not cumulative date for these compounds, the authors should me more modest concerning the activity and add that “may” have therapeutic potential. The same I think concerns and other relevant expressions in the text.

7. The authors should make a revision to the text for some minor grammatical errors.

8. For the Determination of NO production and iNOS expression there are no reference. Furthermore, for iNOS expression quercetin control concentration should be added in order to see whether compound’s 7 activity is comparable or not to quercetin.

9.  “Considering the activity results, compound 7 with relatively high content was selected to detect the iNOS protein expression by western blot” What does high content mean?

I believe that after revision of the manuscript the text could be considered for publication.

Author Response

Reviewer #1:

  1. Response to comment: The “Isolation and identification of compounds” method has been standardized before? Why this system of solvents for extraction was used? In case of another extraction method would have been isolated different groups of compounds? Please add relevant reference in case there are. This is a very important part, since the article concerns Plant Sciences.

Response: We thank the reviewer for his/her interesting questions. First, the “Isolation and identification of compounds” method has not been standardized before. The separation method adopted in this paper is mainly due to the separation habit of Academician Jian Min Yue's group. Petroleum ether, EtOAc and n-butanol extractions are traditional separation methods. Long-term experimental results showed that most of the active compounds existed in the EtOAc-soluble fraction, so we chose this fraction as our isolation object. As reported in the related literature “1. Dawa, Z.M., et al., Chemical constituents of the whole plants of Saussurea medusa. J. Nat. Med. 2009, 63, 327–330. DOI 10.1007/s11418-009-0320-1. 2. Fan, C.Q., et al., Biologically Active Phenols from Saussurea medusa. Bioorg. Med. Chem. 2003, 11, 703708. DOI 10.1016/S0968-0896(02)00470-4. 3. Duan H.Q., et al., Immunosuppressive constituents from Saussurea medusa. Phytochemistry 2002, 59, 85–90. DOI:10.1016/s0031-9422(01) 00429-0”, we believe that another similar or different extraction method can separate different compounds. Some of these reported compounds were also isolated in this study. However, since these compounds are known and not similar in structure, we do not repeat them in this paper. The above mentioned literature has been cited in the article.

  1. Response to comment: “The EtOAc fraction was separated by various column chromatographic methods” Are these the only isolated compounds or there are others that were not isolated and tested?

Response: As mentioned above, there are other compounds isolated from the EtOAc fraction. As we cited in the main text, “1. Cao, J.Y. et al. Saussurenoids A-G, seven new sesquiterpenoids from Saussurea medusa maxim. Tetrahedron 2022, 120, 132850 and 2. Cao, J.Y. et al. Arylnaphthalide lignans from Saussurea medusa and their anti-inflammatory activities. Arab. J. Chem. 2022, 15, 104155. https://doi.org/10.1016/j.arabjc.2022.104155”, a series of new terpenoids and arylnaphthalide lignans were isolated from the EtOAc fraction in our previous study. Some of them exhibited certain anti-inflammatory activities, which prompted us to continue further research on this plant. To the best of our knowledge, this is the first report of 8-O-4' type neolignans in Saussurea medusa. Therefore, we think it is worthy of publication in this reputed journal.

  1. Response to comment: In the supplementary file the authors provide the spectrums of the 3 compounds. What about the other spectrums of the other known compounds?

Response: The other spectra of the known compounds have been supplemented in the supplementary file (Figure S35-S48) according to the reviewer’s comments.

  1. Response to comment: “Along with the new lignans, three known analogues” In this sentence it seems all of them to be known analogues (compounds 4-10), and not the 3 of them.

Response: We are very sorry for the mistake we made, “three” has been changed to “seven”.

  1. Response to comment: The authors should explain how they concluded that these are the structures of compounds 4-10 as they did with compounds 1-3.

Response: The known compounds 4-10 have been analysed and reported in the literature. Additionally, these literatures have been cited in the main text. Moreover, the related spectra of the known compounds have been added to the supplementary material. Considering the length of the paper, we did not repeat these descriptions.

  1. Response to comment: “whole plants of medusa have therapeutic potential in anti-inflammatory diseases” I think that since this is a preliminary study on one target (NO-iNOS) concerning inflammatory processes, and there are not cumulative date for these compounds, the authors should me more modest concerning the activity and add that “may” have therapeutic potential. The same I think concerns and other relevant expressions in the text.

Response: We have made corrections according to the reviewer’s comments. The relevant expressions in the text have also been changed.

  1. Response to comment: The authors should make a revision to the text for some minor grammatical errors.

Response: We have made corrections according to the reviewer’s comments.

  1. Response to comment: For the Determination of NO production and iNOS expression there are no reference. Furthermore, for iNOS expression quercetin control concentration should be added in order to see whether compound’s 7 activity is comparable or not to quercetin.

Response: References for the determination of NO production and iNOS expression have been supplemented in part “Materials and Methods” in the main text.

    In the design of the iNOS expression experiment, we searched related literature and found that the positive control quercetin could definitely inhibit iNOS expression in LPS-induced RAW264.7 cells” 1. Lee, H.N. et al. Anti-inflammatory effect of quercetin and galangin in LPS-stimulated RAW264.7 macrophages and DNCB-induced atopic dermatitis animal models. Int. J. Mol. Med. 2018, 41, 888-898. DOI:10.3892/IJMM.2017.3296. 2. Cui, S. et al. Quercetin inhibits LPS-induced macrophage migration by suppressing the iNOS/FAK/paxillin pathway and modulating the cytoskeleton. Cell adhesion & migration, 2019,13,1-12. DOI:10.1080/19336918.2018.1486142.”. Therefore, we did not repeat the measurement of quercetin. Now, no more compound 7 remains. Moreover, due to the COVID-19 pandemic, we could not obtain the related reagents for a short time. So it is a great pity that we could not complete this experiment. However, we are very grateful for the valuable suggestion put forwards by the reviewer, and we will pay attention to this problem in future experiments.

  1. Response to comment: “Considering the activity results, compound 7 with relatively high content was selected to detect the iNOS protein expression by western blot” What does high content mean?

Response: Compared with other compounds, compound 7 was isolated with a relatively high content of 11 mg. To make the expression clearer and more accurate, we have changed the expression to “high residual amount (11 mg)” in the main text.

Reviewer 2 Report

Hi,

This is a well-written manuscript. I have a few minor comments/concerns:

1. This manuscript includes lots of work to identify the natural products from Saussurea medusa. However, the procedures and results to elucidate the binding condition between iNOS and isolated compounds are not described clearly.

2.     Minor grammatical errors and spelling mistakes needs to be addressed before the final submission.

3.     Authors need to put some effort into explaining polar and non-polar interactions and active site amino acid residues in iNOS.

4.     Discussions need to be elaborated, including anti-inflammatory activities and molecular docking studies.

5.     Overall, this is a well-written manuscript. If the authors include these above-suggested comments, I do not have any problem accepting this paper for publication in this reputed journal.

6.     Please refer to and include the following reference in this manuscript according to the required scenarios.

References.

1.     Shivanagoudra, Siddanagouda R., et al. "Cucurbitane-type compounds from Momordica charantia: Isolation, in vitro antidiabetic, anti-inflammatory activities and in silico modeling approaches." Bioorganic Chemistry 87 (2019): 31-42.

2.     Perera, Wilmer H., et al. "Anti-inflammatory, antidiabetic properties and in silico modeling of cucurbitane-type triterpene glycosides from fruits of an Indian cultivar of Momordica charantia L." Molecules 26.4 (2021): 1038.

Author Response

1. Response to comment: This manuscript includes lots of work to identify the natural products from Saussurea medusa. However, the procedures and results to elucidate the binding condition between iNOS and isolated compounds are not described clearly.

Response: According to the reviewer’s comments, we have supplemented the procedures and results of molecular docking experiments in “3.8. Molecular docking study” and “2.4. Molecular docking studies” in the main text.

2. Response to comment: Minor grammatical errors and spelling mistakes needs to be addressed before the final submission.

Response: We have made corrections according to the reviewer’s comments.

3. Response to comment: Authors need to put some effort into explaining polar and non-polar interactions and active site amino acid residues in iNOS.

Response: According to reviewer’s comments, we have supplemented this content in “2.4. Molecular docking studies” in the main text.

4. Response to comment: Discussions need to be elaborated, including anti-inflammatory activities and molecular docking studies.

Response: According to reviewer’s comments, we have supplemented this content in “2.2. Anti-inflammatory effects and “2.4. Molecular docking studies” in the main text.

5. Response to comment: Overall, this is a well-written manuscript. If the authors include these above-suggested comments, I do not have any problem accepting this paper for publication in this reputed journal.

Response: Thank you very much for the expert's affirmation of this article.

6. Response to comment: Please refer to and include the following reference in this manuscript according to the required scenarios. “1. Shivanagoudra, Siddanagouda R., et al…. Molecules 26.4 (2021): 1038”.

Response: Many thanks to the reviewer for providing these two valuable references. They have been cited in the main text.

Reviewer 3 Report

This is a very interesting study, which is well-designed, performed and presented. The authors have made substantial efforts to isolate and elucidate the structures of the presented compounds, but also to explore their mode of action. Accordingly, I would suggest the publication of this study after addressing the following concerns.

In introduction part

  • In line 60, “NOS” should be changed to “NOSs”
  • In line 61, “, which oxidize” it is suggested to be changed to “by oxidation of”
  • In line 61, there are two isomeric forms of NOSs, so the “Constitutive NOS is” should be changed to “Two constitutive isoforms (cNOS) are”
  • In lines 64 and 65, please cite the relevant studies for the sentence “Inhibiting the activity of the iNOS to block excessive NO production has been considered as a promising strategy for the treatment of inflammatory diseases.”

In results part:

  • Please apply backspace for line 164.
  • In 1H NMR for compounds 1-3, the exchangeable protons were left unassigned.

In Materials and Methods part:

  • The authors have not cited the original studies for “Determination of NO production, Determination of iNOS expression and MTT” assays. Please, as possible cite the original relevant studies.
  • In line 349, the authors stated that “the bands were quantitated by densitometric analysis” without mentioning the software that has been used.
  • In line 352, whenever possible, please refer to the version of docking program.

Author Response

1. Response to comment: In line 60, “NOS” should be changed to “NOSs”

Response: We have made corrections according to the reviewer’s comments.

2. Response to comment: In line 61, “, which oxidize” it is suggested to be changed to “by oxidation of”

Response: We have made corrections according to the reviewer’s comments.

3. Response to comment: In line 61, there are two isomeric forms of NOSs, so the “Constitutive NOS is” should be changed to “Two constitutive isoforms (cNOS) are”

Response: We have made corrections according to the reviewer’s comments.

4. Response to comment: In lines 64 and 65, please cite the relevant studies for the sentence “Inhibiting the activity of the iNOS to block excessive NO production has been considered as a promising strategy for the treatment of inflammatory diseases.”

Response: The relevant reference for the sentence has been cited according to the reviewer’s comments.

5. Response to comment: Please apply backspace for line 164.

Response: We have made corrections according to the reviewer’s comments.

6. Response to comment: In 1H NMR for compounds 1-3, the exchangeable protons were left unassigned.

Response: The 1H NMR (400 MHz) and 13C (125 MHz) data for compound 1 were measured in CDCl3. The probability of observing active hydrogen (the hydroxyl group) is less than 50%. In our study, we did not observe peaks of active hydrogen in the 1H NMR (400 MHz) data of compound 1. The 1H NMR (400 MHz) and 13C (125 MHz) data for compounds 2 and 3 were measured in CD3OD. Active hydrogen will be substituted by deuterium. Therefore, the peak of the active hydrogen cannot be observed.

7. Response to comment: The authors have not cited the original studies for “Determination of NO production, Determination of iNOS expression and MTT” assays. Please, as possible cite the original relevant studies.

Response: The original relevant studies have been cited according to the reviewer’s comments.

8. Response to comment: In line 349, the authors stated that “the bands were quantitated by densitometric analysis” without mentioning the software that has been used.

Response: The software has been added according to the reviewer’s comments.

9. Response to comment: In line 352, whenever possible, please refer to the version of docking program.

Response: The version of docking program has been added according to the reviewer’s comments.

Round 2

Reviewer 1 Report

I would like to thankk the authors for the modifications.

I do not have any more comments.